# Initial Results of Modeling and Improvement of BDS-2/GPS Broadcast Ephemeris Satellite Orbit Based on BP and PSO-BP Neural Networks

**Hanlin Chen** [1]**, Fei Niu** [2]**, Xing Su** [1,3,]*****, Tao Geng** [4] **, Zhimin Liu** [1] **and Qiang Li** [1]

1  College of Geodesy and Geomatics, Shandong University of Science and Technology, Qingdao 266590, China; 201982020022@sdust.edu.cn (H.C.); liuzhimin@sdust.edu.cn (Z.L.); 202083020033@sdust.edu.cn (Q.L.)
2  Beijing Satellite Navigation Center, Beijing 100094, China; niufei009@163.com
3  Key Laboratory of Geomatics and Digital Technology of Shandong Province, Shandong University of Science and Technology, Qingdao 266590, China
4  GNSS Research Center, Wuhan University, 129 Luoyu Road, Wuhan 430079, China; gt_gengtao@whu.edu.cn
*  Correspondence: suxing@sdust.edu.cn; Tel.: +86-532-8605-8006

**Abstract:** With the rapid development and gradual perfection of GNSS in recent years, improving the real-time service performance of GNSS has become a research hotspot. In GNSS single-point positioning, broadcast ephemeris is used to provide a space–time reference. However, the orbit parameters of broadcast ephemeris have meter-level errors, and no mathematical model can simulate the variation of this, which restricts the real-time positioning accuracy of GNSS. Based on this research background, this paper uses a BP (Back Propagation) neural network and a PSO (Particle Swarm Optimization)–BP neural network to model the variation in the orbit error of GPS and BDS broadcast ephemeris to improve the accuracy of broadcast ephemeris. The experimental results showed that the two neural network models in GPS can model the broadcast ephemeris orbit errors, and the results of the two models were roughly the same. The one-day and three-day improvement rates of RMS(3D) were 30–50%, but the PSO–BP neural network model was better able to model the trend of errors and effectively improve the broadcast ephemeris orbit accuracy. In BDS, both of the neural network models were able to model the broadcast ephemeris orbit errors; however, the PSO–BP neural network model results were better than those of the BP neural network. In the GEO satellite outcome of the PSO–BP neural network, the STD and RMS of the orbit error in three directions were reduced by 20–70%, with a 20–30% improvement over the BP neural network results. The IGSO satellite results showed that the PSO–BP neural network model output accuracy of the along- and radial-track directions experienced a 70–80% improvement in one and three days. The one- and three-day RMS(3D) of the MEO satellites showed that the PSO–BP neural network has a greater ability to resist gross errors than that of the BP neural network for modeling the changing trend of the broadcast ephemeris orbit errors. These results demonstrate that using neural networks to model the orbit error of broadcast ephemeris is of great significance to improving the orbit accuracy of broadcast ephemeris.

**Keywords:** satellite orbit error; neural network; PSO; AODE; BDS

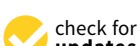



## 1. Introduction

With the construction of GNSS in recent years, single-point positioning as one of the main functions of GNSS has become a research hotspot [1–3]. In single-point positioning and navigation, the accuracy of broadcast ephemeris is very important [4,5]. Currently, the accuracy of GPS and BDS is evaluated from multiple aspects based on MGEX broadcast ephemeris and precision ephemeris. The overall orbit accuracy of GPS is better than 0.3 m, the clock error of the RMS is better than 0.4 m, and the SISRE (Signal-In-Space Range Error) of the RMS is better than 0.5 m; for BDS, the overall orbit accuracy of BDS-2 is better than

1.0 m, the RMS of the clock difference is better than 0.6 m, and the RMS of SISRE is better than 1 m [6]. However, in [7], the broadcast ephemeris accuracy of 14 BDS-2 satellites was analyzed. The broadcast ephemeris accuracy of the IGSO and MEO satellites was shown to be close to that of GPS, with the three-dimensional accuracy basically within 3 m; meanwhile, the GEO satellite accuracy was worse, with the three-dimensional accuracy being approximately 10 m. With the continuous development of BDS, analysis based on the precision orbit products provided by the iGMAS Product Center from 2014 to 2017 has shown that the orbit accuracy of the BDS IGSO and MEO satellites has remained stable, the three-dimensional accuracy of the RMS is basically 1 m, and the URE (User Range Error,) is mostly distributed between 0.5 and 1 m; meanwhile, the three-dimensional accuracy of GEO satellites has improved from more than 10 m to approximately 3 m, URE has reduced from nearly 2 m to about 1 m, and the overall three-dimensional accuracy of the system has improved from approximately 6 m to around 2 m [8]. In [9], the orbit accuracy and clock error accuracy of the BDS GEO, IGSO, and MEO satellites, as well as the overall accuracy of SISRE, were analyzed and compared in terms of their short- and long-term trends, showing that the orbit accuracy of the GEO satellites is significantly lower than that of the IGSO and MEO satellites. In a comparison of the three directions, the radial-track direction accuracy was the highest—approximately 0.5 m; the cross-track direction came second, with the GEO satellites better than 3 m, and the MEO satellite better than 1.5 m; along-track direction was the worst direction—greater than 2 m. The orbit accuracy of the BDS-3 broadcast ephemeris in along-, cross-, and radial-track directions is 0.30, 0.26, and 0.07 m, respectively, which is significantly better than that of BDS-2 [10]. Therefore, in the joint positioning of the BDS-2 and BDS-3 systems, there is a problem of weight determination between them. However, there is no better solution for the weight matrix determination of the joint positioning of BDS-2 and BDS-3; therefore, this article focuses on modeling the satellite orbit errors of BDS-2 broadcast ephemeris to improve its accuracy. By increasing the accuracy of the BDS-2 broadcast ephemeris to be comparable to that of BDS-3, the problem of determining the weight of the two systems in joint positioning can be eliminated.

Due to the lower orbit accuracy of broadcast ephemeris satellites, satellite orbit predictions are usually carried out using precise orbit and dynamic models. In the trajectory prediction of the dynamic model, the error increases with the length of the forecast arc. Therefore, the neural network model was used to learn the trajectory prediction error of the kinetic model and to provide compensation for the prediction, thereby improving the accuracy of the kinetic model orbit. In [11], the neural network model was introduced as compensation on the basis of the original dynamic model, and the characteristics of the long-term prediction error in the GPS satellite dynamic model were analyzed. A simulation test of the GPS satellite proved that the improved effect of the hybrid model is better in the 15–40-day forecast. In [12], the BP neural network and LSTM (Long Short-Term Memory) neural network models were used to compensate for the orbit errors of the dynamic model prediction. The results showed that the LSTM model can learn the error characteristics more completely than the BP neural network. In improving the short-, medium-, and long-term error compensation, the LSTM neural network can obviously compensate for the orbit error of the GEO, IGSO, and MEO satellites compared to the BP neural network. In [13], a neural network model was established on the basis of the dynamic model to predict the orbit of different BDS satellites, and the conclusion showed that the improvement effect differs at different initial moments. The BDS satellite orbit accuracy of the forecast 15 days improved from 318 to 19 m, and the orbit accuracy of the 30-day forecast improved from 1757 to 49 m. The 15- and 30-day orbit improvements were predicted to be 41–80% and 32–88%, respectively.

At present, the characteristics of real-time broadcast of broadcast ephemeris and the variation of the broadcast ephemeris orbit error have not yet been described by an accurate mathematical model. In [14], only the neural network model was used to predict the orbit error of the satellite in the broadcast ephemeris under the geocentric coordinate

system. It is impossible to analyze the correlation between the force state of a satellite and the orbit characteristics, so there is less research on modeling the orbit errors of a broadcast ephemeris satellite and the improvement of its satellite orbit accuracy. Based on this research background, this article uses MEGX precision ephemeris as a reference, together with satellite coordinates, velocities, and corresponding epoch time in the broadcast ephemeris as training samples to model BP and PSO–BP neural network models. After training, we input the test data samples into the two neural network models and obtained the broadcast ephemeris satellite orbit errors of the model output. Finally, we removed the model outputs from the actual errors to improve the orbit accuracy of the broadcast ephemeris.

## 2. Materials and Methods

To model the orbit error of broadcast ephemeris, we began with analysis of the variation of the orbit error. Because the variation of the orbit error has no obvious regularity and the traditional method has no advantage in modeling details, we propose the use of BP and PSO–BP neural networks to model the orbit errors of broadcast ephemeris.

### 2.1. Orbit Error of Broadcast Ephemeris

The orbit errors of broadcast ephemeris are referenced by the precise orbit products of MGEX. MGEX broadcast ephemeris is downloaded from the IGS ftp, among which BDS broadcast navigation message broadcasts were set to one every hour, in which GPS broadcast navigation message broadcasts were set to one every two hours. We calculated the orbit errors of the broadcast ephemeris of all BDS-2 and GPS satellites for the 35th to the 65th days of 2020. To ensure the reliability and rigor of the experiment, the data from the 50th to the 59th days of 2020, which have continuous and complete data resulting from more satellites, were selected for the experiments.

After all necessary datasets were collected, the satellite position coordinates in the earth-fixed coordinate system of broadcast ephemeris were compared to the precise satellite position coordinates of the MGEX precise orbit product in the corresponding epoch to obtain the broadcast ephemeris orbit errors. In particular, as the orbit error results are the direct differences between the broadcast ephemeris and precise products, the differences in the coordinate systems are included. The error was quite large, thus facilitating subsequent data processing.

According to the orbit errors of the BDS-2 satellite of broadcast ephemeris from the 50th to the 59th day of 2020, shown in Figure 1, the three satellite orbit errors in the cross-track and radial-track directions were within $\pm 4$ m, while those in the along-track direction were within $\pm 10$ m. The errors in the three directions all showed periodic variation, but the error noise was included in the change. It is impossible to use the existing mathematical model to accurately model this changing trend. Figure 2 shows the changes in the orbit errors of the broadcast ephemeris of some GPS satellites. Figure 2a shows the variation in the along-, cross-, and radial-track directions of the G03, G13, and G16 satellites, while Figure 2b shows the variation in the along-, cross-, and radial-track directions of the G20, G24, and G30 satellites. The six GPS satellites showed the most severe error trend in the along-track direction, with a larger error value than in the other two directions.

Figure 2 shows that the orbit error of the six GPS satellites in the along-, cross-, and radial-track directions were within $\pm 4$ m. It also shows the same regular change including error noise. Its future change trend cannot be accurately modeled, so we used the BP and PSO–BP neural networks to model the orbit errors of the broadcast ephemeris.

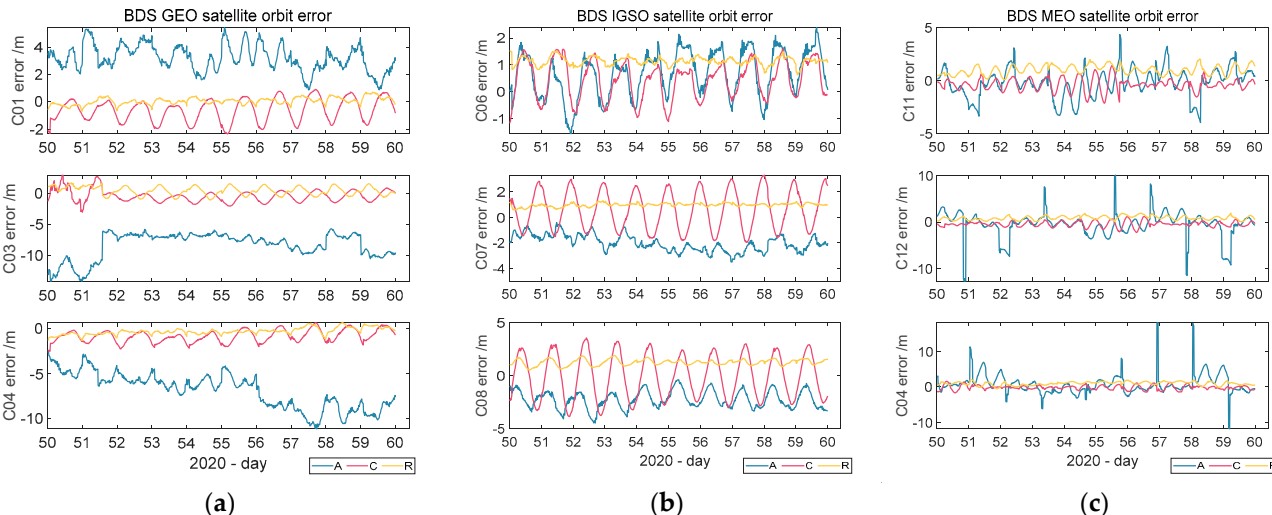

**Figure 1.** Variation in the orbit errors of BDS satellites in different orbits: (**a**) BDS GEO satellite orbit errors of C01\C03\C04; (**b**) BDS IGSO satellite orbit errors of C06\C07\C08; (**c**) BDS MEO satellite orbit errors of C11\C12\C14.

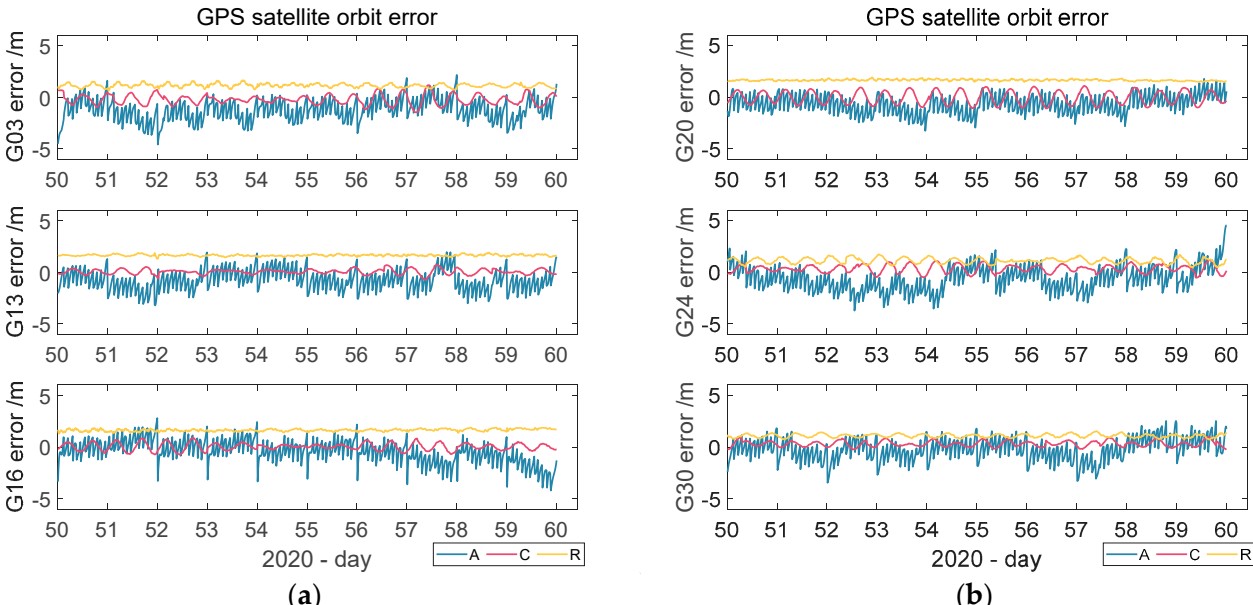

**Figure 2.** Variation of the orbit errors of the GPS satellites: (**a**) G03/G13/G16 orbit error; (**b**) G20/G24/G30 orbit error.

### 2.2. Impact of AODE of BDS

By counting the AODE (Age of Data and Ephemeris) data in the broadcast ephemeris of the BDS satellites in different orbits from 50 to 59 days, Figure 3 shows that the AODE changed and the extreme values of different orbit satellites were different.

Because the BDS ground stations are located in China, the AODE data value of the BDS satellites was larger than that of the GPS satellites, and the influence of the satellite orbit errors was also at a disadvantage.

In BDS, the AODE was the largest among the MEO satellites, and its maximum value was close to 20 h. In the IGSO satellites, the maximum AODE was only 6 h, while the GEO satellites are geostationary orbit satellites, meaning their AODEs were close to 1 h.

The unique IODE time length from the broadcast ephemeris indicates that the change of the satellite AODE value is always less than or equal to 2 h, as shown in Figure 4, which has little impact on the satellite orbit accuracy of the broadcast ephemeris.

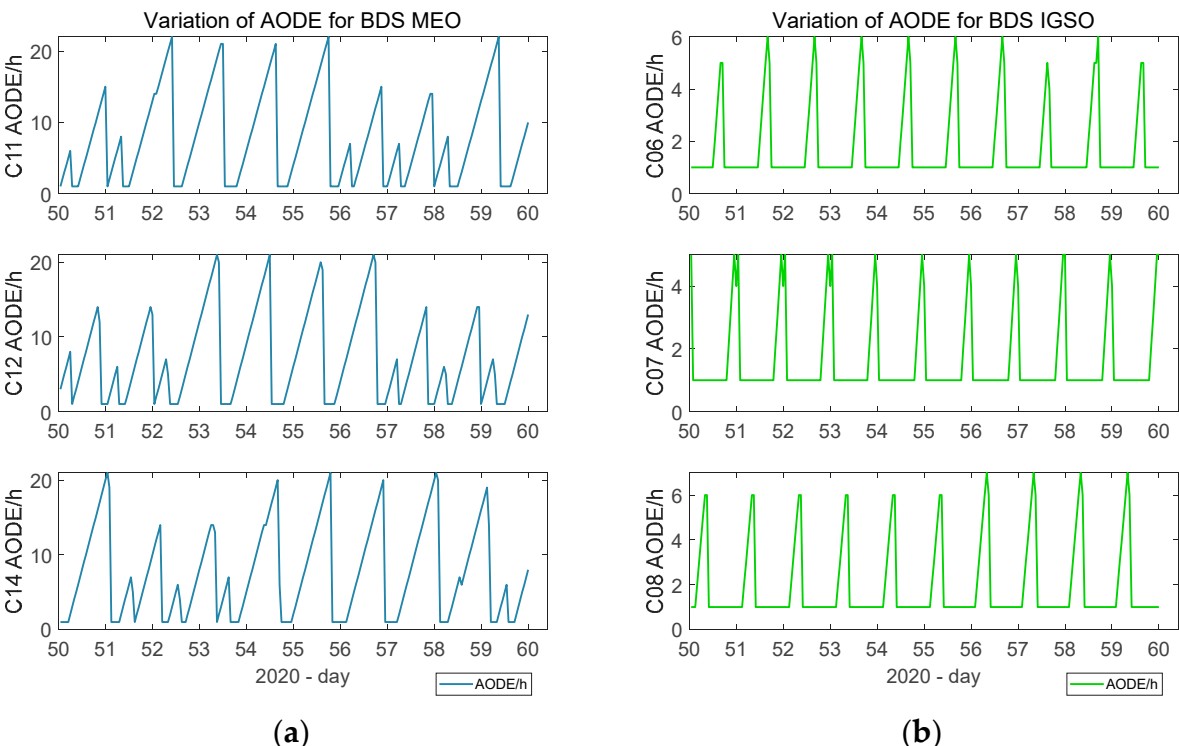

**Figure 3.** Variation in the AODE for the BDS satellite: (**a**) BDS MEO satellite AODE; (**b**) BDS IGSO satellite AODE.

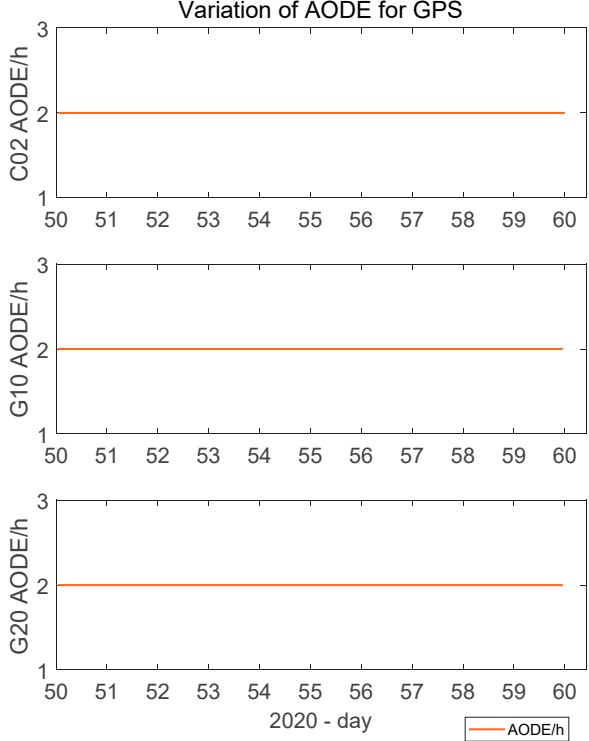

**Figure 4.** Variation in the AODE for GPS satellites.

### 2.3. Model of BP Neural Network

Establishing a mathematical model is the basis of satellite navigation data processing. At present, the mathematical models in the field of expressing satellite orbit and clock offset mainly include the elliptical satellite orbit model, the secondary satellite clock model, and so on. This kind of mathematical model, which can be expressed by the function of determining causality, is called a deterministic model. However, in the practice of satellite navigation data processing, it has also been found that there are some objectively regular phenomena beyond the deterministic model expression capabilities currently constructed, which cannot be described by simple mathematical formula, nor can they be modeled by the traditional method, especially in BDS-2. This kind of mathematical model, which cannot be expressed by the function of determining causality, is called an uncertainty model.

The nonlinear mapping ability of BP neural networks makes these networks essentially realize a mapping function from input to output. Mathematical theory has proven that a three-layer neural network can approximate any uncertain model with arbitrary precision. This makes it particularly suitable for solving problems with complex internal mechanisms. That is, BP neural networks have a strong capability to model uncertainty models [15].

BP neural network algorithms include two processes: forward propagation of signals and back propagation of errors. When propagating forward, the signal propagates from the input layer to the output layer through the hidden layer. If the results of the output layer do not meet the expectations, then the propagated signal transfers to back propagation. Errors return in the original path and adjust the weights and thresholds of the network to minimize the sum of squared errors of the network; then, the errors transfer to forward propagation. This process repeats until the errors are less than the set value.

Figure 5 presents the main process of BP neural networks.

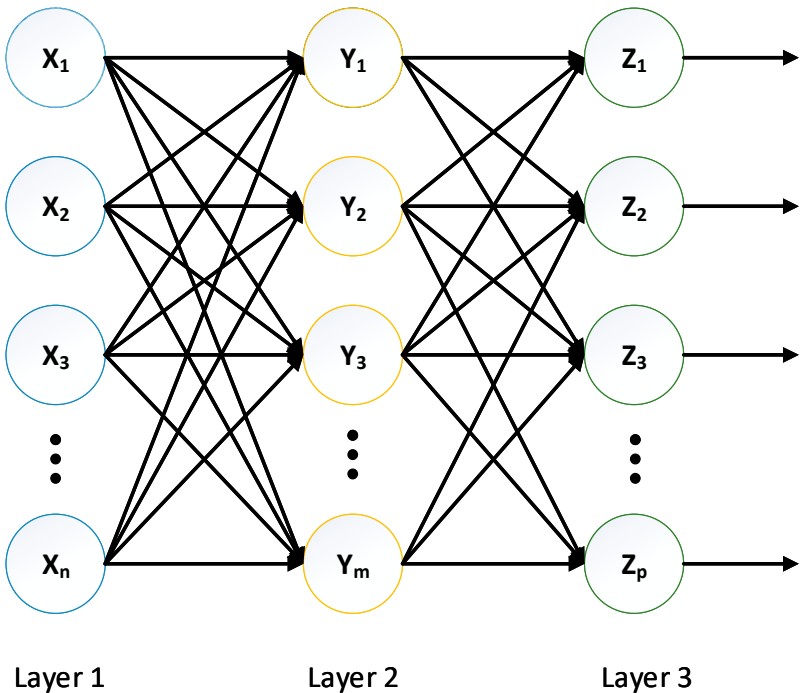

**Figure 5.** Three-layer BP neural network model error propagation. Layers 1, 2, and 3 are the input, hidden, and output layers, respectively.

Currently, in related research, it is inconclusive whether increasing the number of hidden layers can reduce the network error, but it undoubtedly complicates the network structure and greatly increases the network training time and data occupation space. Therefore, a three-layer network with a single hidden layer is generally preferred in design, and the number of hidden layer nodes is changed by the trial and error method to obtain

the network structure with the smallest error. These lections of the number of hidden layer nodes also have a great influence on the performance of the neural network model, which is the direct cause of "overfitting" in training. At present, the formulas given in most research are for any number of samples, and most are for the most disadvantageous situation, so they are not recommended for practical engineering applications. Based on the basic principle that the network structure should be as compact as possible on the premise of meeting the accuracy requirements, comprehensively considering the complexity of the network structure, the size of the error, and the number of samples, this model has been trained many times using the node deletion method. After performance comparison, the optimal number of hidden layer nodes was determined to be five [15,16].

In this research, the main purpose was to model the orbit errors of broadcast ephemeris. In broadcast ephemeris, the results related to the orbit error that can be calculated are coordinates and velocities of the satellites in the three directions in the earth-fixed coordinate system, and there is an indeterminate periodicity in the time series of the broadcast ephemeris orbit errors, noted in Section 2.1. Based on these, we decided that the BP neural network model input layer has seven nodes, which are the coordinates and velocities of the satellites in the three directions in the earth-fixed coordinate system and corresponding epoch time. The hidden layer has five nodes, there are three nodes in the output layer, the orbit errors of the along-, cross-, and radial-track directions.

*2.4. PSO–BP Neural Network*

Particle swarm optimization (PSO) can be used to optimize the weights and thresholds of the BP neural network, which can effectively prevent the training from falling into the local optimal situation, meaning that the error of the neural network training can be minimized [17,18].

PSO is a kind of evolutionary algorithm, which is designed by simulating the predation behavior of birds. Starting from the random solution, the optimal solution is searched through iteration, and the quality of the solution is evaluated by fitness. Here, let us take an example to understand the algorithm in depth: Suppose there is a flock of birds and food is placed somewhere on an island; the birds do not know where the food is, but only know the distance to the food. These are such smart birds that they search around the birds that are currently closest to the food so that they can find food sooner.

PSO is initialized as a group of random particles (random solution), and then the optimal solution is found through iteration. In each iteration, the particle keeps up with itself by tracking two "extremums". The first is the optimal solution *pbest* found by the particle itself; the other extreme is the optimal solution currently found by the entire population, the global extreme *gbest*. The particle uses the following formula to update its speed and position [19–21]:

Speed conversion formula:

$$v_{i+1} = w \times v_i + c_1 \times rand_1 \times (pbest_i - x_i) + c_2 \times rand_2 \times (gbest_i - x_i) \qquad (1)$$

Position transformation formula:

$$x_i = x_i + v_{i+1} \qquad (2)$$

where $w$ is the inertia factor, generally 1; $c_1$ and $c_2$ are the learning factors, generally 2; $rand_1$ and $rand_2$ are random numbers between (0,1); $v_i$ and $x_i$ represent the velocity and position of the *i*th dimension of the particle; $pbest_i$ and $gbest_i$ represent the value of the *i*th dimension at the best position of a particle and the value of the *i*th dimension at the best position of the entire population. These two formulas are updated for a certain dimension of particles. For each dimension of the particles, the formula must be used to update.

According to the above basic process of the PSO algorithm, the combination process of the PSO algorithm and the BP neural network follows [21]:

1. Generate a group of particle swarm to initialize the weight and threshold on each node of the BP neural network.

2. Calculate the fitness of PSO by the function of the BP neural network prediction error.

3. Use the PSO calculated particle swarm in the BP neural network and record the results of the test sample data.

4. If the PSO algorithm reaches the maximum number of iterations, the trained particle swarm parameters are output as the weights and thresholds of the BP neural network. Otherwise, the particle swarm parameters are renewed based on the process of the PSO algorithm, returning to step 1 to calculate the new result.

Finally, the particle swarm optimization obtained by the PSO algorithm is applied to the optimization of the weights and thresholds of the BP neural network to replace the iterative learning process of this network and to improve the learning efficiency. The input and output parameters and the number of hidden layer nodes of the PSO–BP neural network model are the same as those of the BP neural network. The particle swarm size of the PSO algorithm is 200, the maximum number of iterations is set to five, the learning factor is $c_2 = c_2 = 2$, and the inertia weight is 0.95.

## 2.5. Experiment Process

At first, according to the calculated one-month broadcast ephemeris orbit errors, the data from the 50th to the 59th days of 2020 were selected for the experiment after excluding the satellites with missing data. We obtained the broadcast ephemeris and precision ephemeris files from the 50th to the 56th days of 2020, and calculated the coordinates and velocities of the three directions of the satellite under the earth-fixed coordinate system through broadcast ephemeris. Second, using precise orbit products, the satellite coordinates were used as reference values to calculate the orbit errors of the corresponding epoch broadcast ephemeris satellites in the three directions of along-, cross-, and radial-track. Then, the BP and PSO–BP neural network models were established, and the broadcast ephemeris satellite coordinates and velocities of the three directions of the earth-fixed coordinate system and the corresponding epoch time from the 50th to the 56th days of 2020 were used as the input training data. The orbit errors of the along-, cross-, and radial-track directions of the satellite orbit were used as the output training data to train the two neural network models. After this training, the broadcast ephemeris satellite coordinates and velocities of the three directions of the earth-fixed coordinate system and the epoch time on the 57th or 57th–59th days were imported into the two neural network models as input test data. Finally, the output of the satellite orbit's along-, cross-, and radial-track directions of the 57th or 57th–59th days by the two neural network models was obtained. The errors were compared to the satellite orbit error in the actual broadcast ephemeris to analyze the modeling effect of the two neural network models and the compensation results of the satellite orbit error of the broadcast ephemeris. Figure 6 describes the flow of the entire experimental method.

Finally, we used RMS(3D) to evaluate the accuracy of all satellite model outcomes; the formula of RMS(3D) is:

$$RMS_{i(3D)} = \sqrt{RMS^2_{i(Along-track)} + RMS^2_{i(Cross-track)} + RMS^2_{i(Radial-track)}} \qquad (3)$$

where $RMS^2_{i(Along-track)}$, $RMS^2_{i(Cross-track)}$, and $RMS^2_{i(Radial-track)}$ are the orbit error of the *RMS* in the three directions of the satellite.

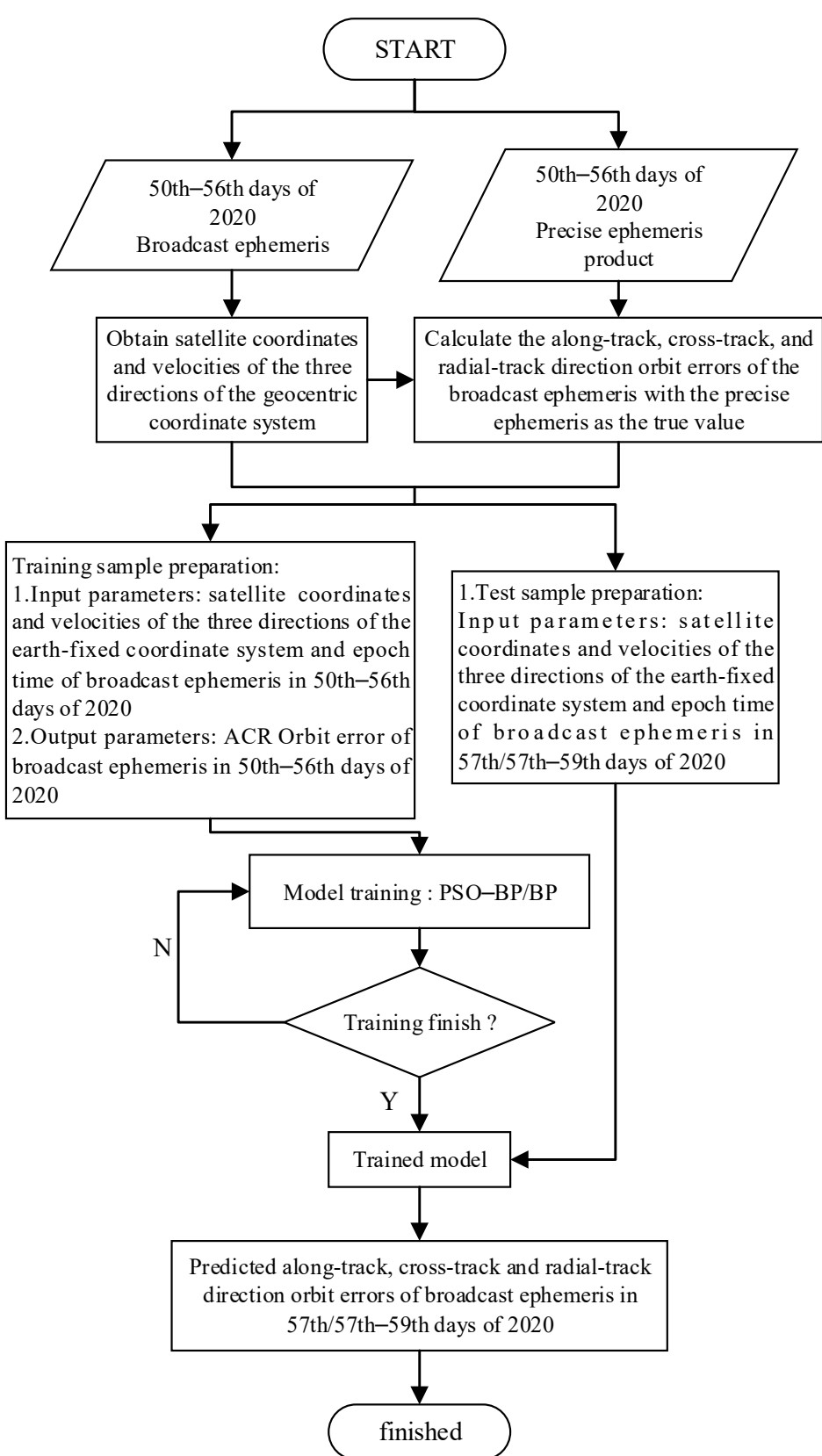

**Figure 6.** Process of the experiment.

## 3. Results

In this section, we first modeled BDS and GPS satellite orbit errors of broadcast ephemeris using the BP and PSO–BP neural networks. Then, the differences in output of the results of the two models in the different satellite orbits are reported.

### 3.1. BDS Satellite Orbit Error Model Output

BDS has three different orbits, i.e., GEO, IGSO, and MEO. There are different rules of variation in these different orbits, so we show the satellite model output of the orbit errors of each orbit to analyze the details of the PSO–BP and BP neural network models.

#### 3.1.1. GEO Orbit

Because the GEO satellite orbit is a geosynchronous orbit, it needs continuous orbit maintenance, so the GEO satellite orbit error is relatively large.

Figure 7a shows the one-day model output difference. According to the curve changes, the outcome of the PSO–BP neural network in the along-track direction at 2–8 h, the cross-track direction at 12–18 h, and the radial-track direction 2–6 h, which were closer to 0 m, were significantly smaller than those of the BP neural network. In the remainder of the time period, the modeling effect of the PSO–BP and BP neural networks were relatively close. Figure 7b shows the three-day model output differences. The outcomes of the two models in the along- and cross-track directions were similar to that in one day. In the radial-track direction, the modeling effect of the PSO–BP neural network was more stable than that of the BP neural network.

In addition, the STD and RMS of the prediction results of BP and PSO–BP neural network models in Table 1 further prove that the one-day and three-day PSO–BP prediction effects are better; additionally, the one-day PSO–BP neural network demonstrated a STD in three directions compared to the BP neural network prediction results, and the RMS reduced by 20–30%. Both models reduced most of the orbit errors; the STD and RMS of the broadcast ephemeris in the cross-track direction were reduced by two orders of magnitude, while the along- and radial-track directions were reduced by one order. Moreover, the three-day STD and RMS of the PSO–BP neural network in the along-track direction compared to the three-day STD and RMS of the BP neural network prediction were reduced by 20–30%.

#### 3.1.2. IGSO Orbit

Because the IGSO broadcast ephemeris orbit error variation was different to that of the GEO satellites, the model results obtained were also different to those of the GEO satellites.

Figure 8 shows that the IGSO modeling outcome graph; specifically, Figure 8a shows that the one-day output of the PSO–BP neural network model was better than that of the BP neural network in the along- and cross-track directions, with its outcome closer to 0 m, which can basically eliminate the broadcast ephemeris through modeling orbit errors. Meanwhile, Figure 8b shows that the three-day fluctuation in the outcome of the BP neural network became more violent than that of the PSO–BP neural network in the three directions. The accuracy of the model prediction results in Table 2 shows that the one-day output results of the IGSO satellite PSO–BP neural network model in the along- and cross-track directions were 70–80% higher than those of the BP neural network in terms of the STD and RMS, and the results of the two models in the cross-track direction were relatively close. The three-day results in Table 2 show that the RMS of the outcome of the two models in three directions demonstrated poorer accuracy than the one-day results, but the precision was still better than the broadcast ephemeris.

**Table 1.** C03 satellite broadcast ephemeris and model-refined ephemeris orbit errors.

| Days | Direction | Along-Track | | | Cross-Track | | | Radial-Track | | |
|------|-----------|-------|-------|-------|-------|-------|-------|-------|-------|-------|
| | Model | Mean/m | STD/m | RMS/m | Mean/m | STD/m | RMS/m | Mean/m | STD/m | RMS/m |
| | Real | −9.06 | 0.47 | 9.07 | −0.43 | 0.78 | 0.89 | 0.26 | 0.61 | 0.66 |
| 1 d | BP | 0.18 | 1.92 | 1.91 | 0.10 | 0.28 | 0.29 | 0.01 | 0.40 | 0.40 |
| | PSO–BP | −0.73 | 0.46 | 0.87 | −0.05 | 0.14 | 0.15 | −0.02 | 0.10 | 0.10 |
| | Real | −8.48 | 1.42 | 8.60 | −0.39 | 0.77 | 0.86 | 0.32 | 0.62 | 0.70 |
| 3 d | BP | −1.43 | 1.45 | 2.03 | 0.52 | 0.29 | 0.60 | −0.09 | 0.19 | 0.21 |
| | PSO–BP | −0.38 | 1.41 | 1.46 | 0.55 | 0.29 | 0.62 | −0.01 | 0.22 | 0.22 |

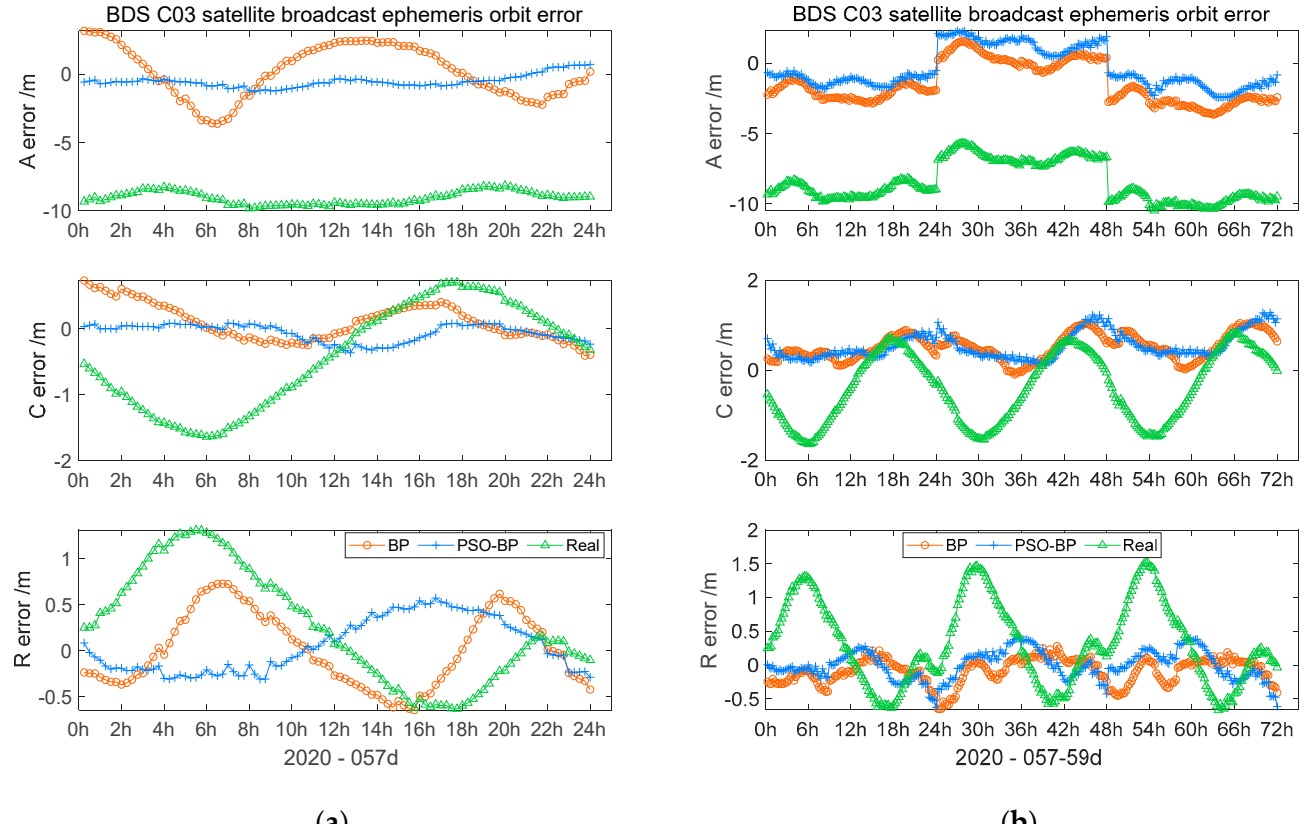

(**a**)                    (**b**)

**Figure 7.** BDS C03 satellite orbit modeling outcome: (**a**) BDS C11 satellite orbit modeling outcome on the 57th day of 2020;
(**b**) BDS C11 satellite orbit modeling outcome on the 57th–59th days of 2020.

**Table 2.** C08 satellite broadcast ephemeris and model-refined ephemeris orbit errors.

| Days | Direction | Along-Track | | | Cross-Track | | | Radial-Track | | |
|------|-----------|-------|-------|-------|-------|-------|-------|-------|-------|-------|
| | Model | Mean/m | STD/m | RMS/m | Mean/m | STD/m | RMS/m | Mean/m | STD/m | RMS/m |
| | Real | −2.33 | 0.74 | 2.44 | −0.03 | 2.01 | 2.00 | 1.27 | 0.19 | 1.29 |
| 1 d | BP | −0.28 | 0.15 | 0.31 | 0.08 | 0.19 | 0.21 | 0.04 | 0.08 | 0.09 |
| | PSO–BP | −0.27 | 0.15 | 0.30 | 0.10 | 0.18 | 0.21 | 0.03 | 0.07 | 0.08 |
| | Real | −2.26 | 0.80 | 2.39 | 0.03 | 1.93 | 1.93 | 1.32 | 0.22 | 1.33 |
| 3 d | BP | −1.27 | 0.63 | 1.41 | 0.52 | 0.76 | 0.92 | 0.35 | 0.37 | 0.51 |
| | PSO–BP | −0.47 | 0.24 | 0.53 | 0.06 | 0.33 | 0.33 | −0.08 | 0.17 | 0.19 |

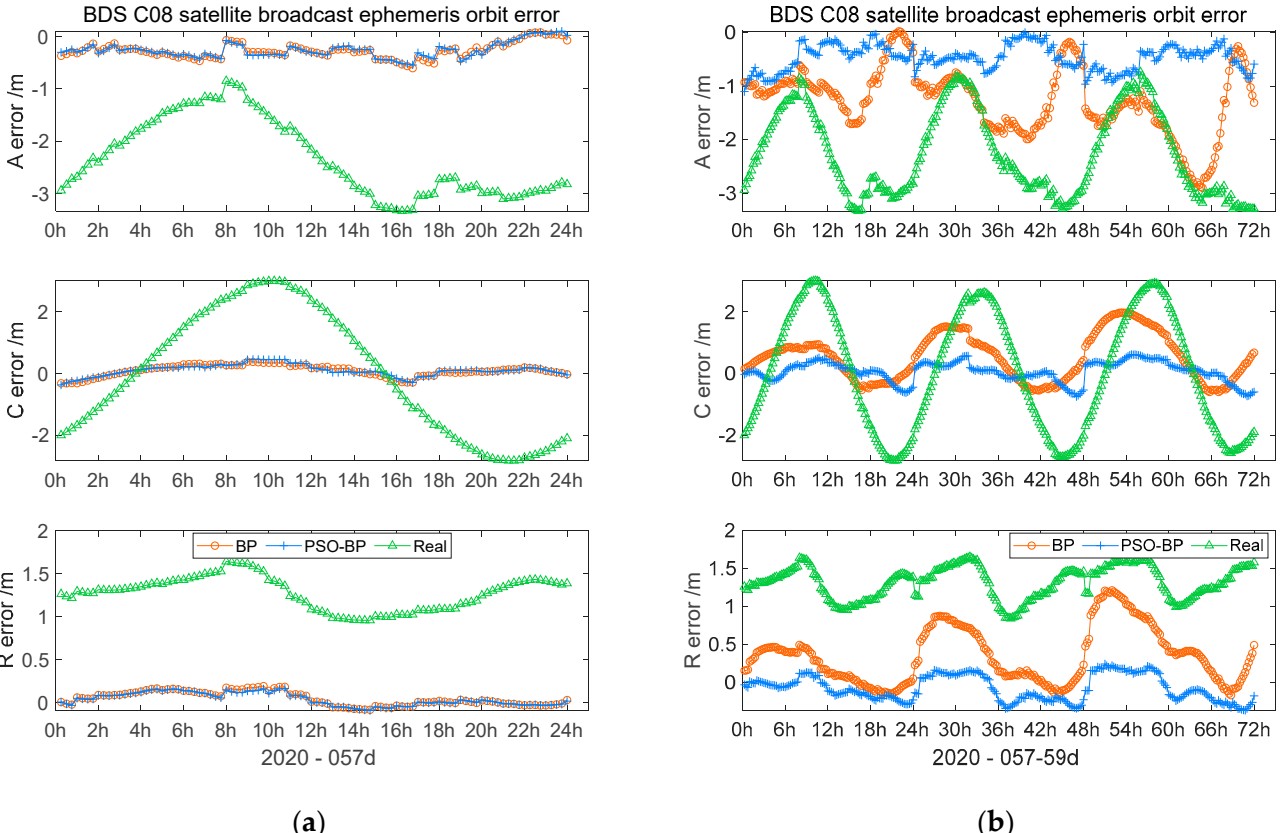

**Figure 8.** BDS C08 satellite orbit modeling outcome: (**a**) BDS C08 satellite orbit modeling outcome on the 57th day of 2020; (**b**) BDS C08 satellite orbit modeling outcome on the 57th–59th days of 2020.

### 3.1.3. MEO Orbit

In the training of the neural network model, due to the influence of the BDS MEO satellite AODE, the satellite orbit error changes are more complicated. Therefore, there was a large deviation in the model output of the MEO satellite orbit error and the output shows that the prediction results of the two neural network models were similar.

In Figure 9a, the MEO satellite one-day modeling outcome graph shows that the all-day prediction effect of the PSO–BP neural network in the along- and cross-directions was similar to that of the BP neural network, while the PSO–BP neural network model results in the radial-track direction having smaller residuals and better accuracy compensation. Figure 9b displays the three-day results; the variation of the curve in the along- and cross-track directions is roughly the same as that of the one-day results. In the radial-track direction, the PSO–BP model output curve is closer to 0 m.

In Table 3, the one-day results of the PSO–BP neural network model in the along- and cross-track directions are similar to those of the BP neural network in terms of the STD and RMS. The modeling effect of the BP neural network in the along-track direction is slightly better than that of PSO–BP neural network, but with unobvious advantages. Compared to the BP neural network, the modeling accuracy of the PSO–BP neural network in the radial-track direction was improved by 30%. Meanwhile, the three-day results of the three directions in Table 3 show RMS and STD of two models became larger than that of the one-day results.

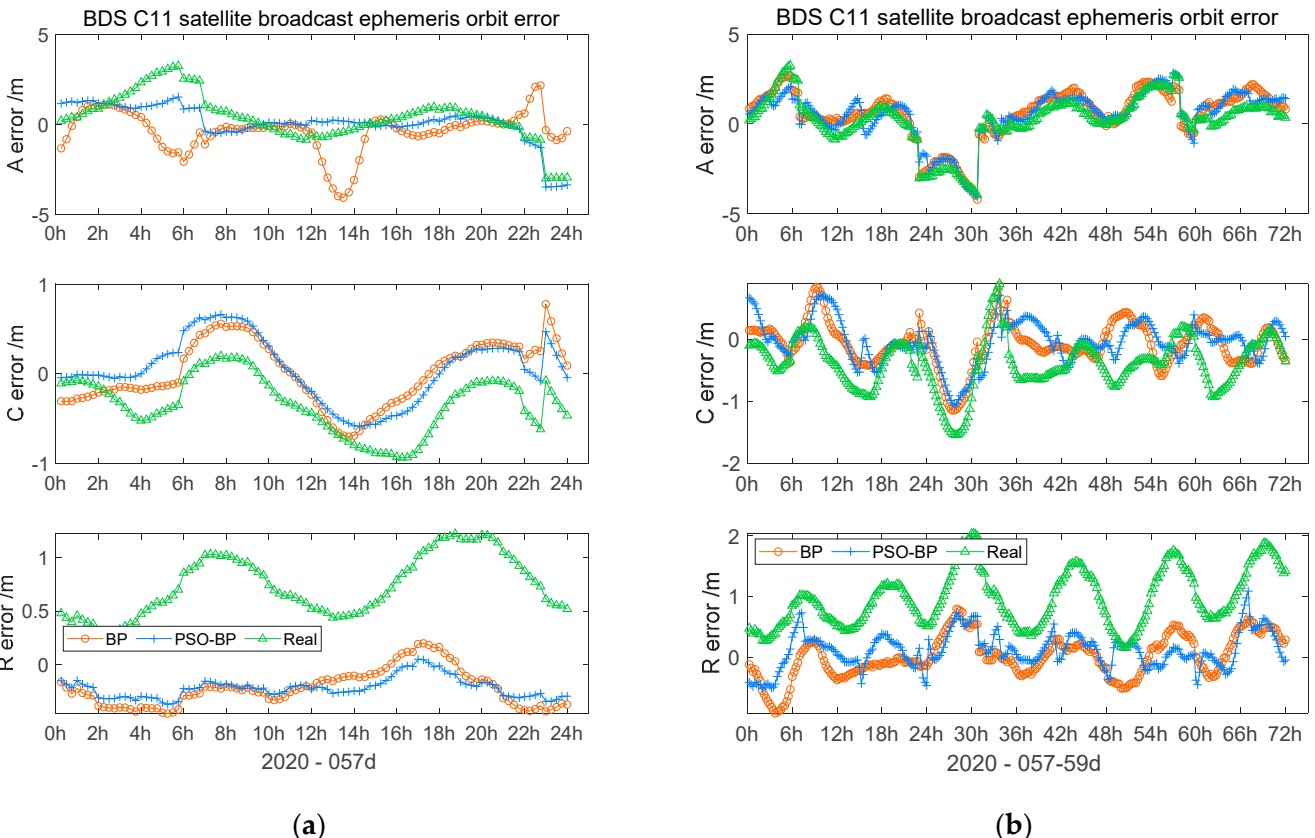

**Figure 9.** BDS C11 satellite orbit modeling outcome: (**a**) BDS C11 satellite orbit modeling outcome on the 57th day of 2020; (**b**) BDS C11 satellite orbit modeling outcome on the 57th–59th days of 2020.

**Table 3.** C11 satellite broadcast ephemeris and model-refined ephemeris orbit errors.

| Days | Direction | Along-Track | | | Cross-Track | | | Radial-Track | | |
|------|-----------|-------------|---|---|-------------|---|---|--------------|---|---|
| | Model | Mean/m | STD/m | RMS/m | Mean/m | STD/m | RMS/m | Mean/m | STD/m | RMS/m |
| | Real | 0.40 | 1.30 | 1.35 | −0.33 | 0.32 | 0.46 | 0.74 | 0.28 | 0.79 |
| 1 d | BP | −0.38 | 1.14 | 1.20 | 0.01 | 0.35 | 0.35 | −0.21 | 0.17 | 0.27 |
| | PSO–BP | 0.11 | 1.03 | 1.03 | 0.05 | 0.35 | 0.35 | −0.21 | 0.09 | 0.23 |
| | Real | 0.22 | 1.40 | 1.41 | −0.39 | 0.42 | 0.57 | 0.97 | 0.47 | 1.08 |
| 3 d | BP | 0.56 | 1.40 | 1.51 | −0.08 | 0.36 | 0.37 | 0.01 | 0.35 | 0.35 |
| | PSO–BP | 0.52 | 1.33 | 1.43 | −0.04 | 0.35 | 0.35 | 0.12 | 0.29 | 0.32 |

Table 4 shows the results of the BDS-2 satellites with no missing data. In the one- and three-day results, the model output of more than 70% of the satellites can compensate for the broadcast ephemeris orbit errors. In the one-day results for the GEO satellites, except for the C01 satellite, the improvement rate of the PSO–BP neural network reached 70–90% and the accuracy improved by 20–30% compared to the BP neural network results; meanwhile, the orbit accuracy of the IGSO satellites improved by 60–90% for the PSO–BP neural network, which is roughly the same as that of the BP neural network. Except for C11, the BP neural network outcomes of the C12 and C14 satellites of the MEO satellites could not improve the accuracy of the broadcast ephemeris orbit. In the three-day results of the two models, except for the MEO satellite, the compensation effect of the other satellites decreased because the output time of the model expanded and the ability of the cumulative control error reduced satellite orbit error model output.

**Table 4.** BDS satellite broadcast ephemeris and model-refined ephemeris orbit errors RMS(3D) and improvement rate.

| PRN | RMS(3D)/m (1 Day) | | | Improvement Rate (1 Day) | | RMS(3D)/m (3 Days) | | | Improvement Rate (3 Days) | |
|-----|------|------|--------|------|--------|------|------|--------|------|--------|
| | Real | BP | PSO–BP | BP | PSO–BP | Real | BP | PSO–BP | BP | PSO–BP |
| C01 | 2.57 | 1.37 | 1.57 | 47% | 39% | 2.88 | 1.79 | 1.67 | 38% | 42% |
| C03 | 9.14 | 1.98 | 0.77 | 78% | 92% | 8.67 | 2.13 | 1.60 | 75% | 82% |
| C04 | 9.77 | 3.70 | 2.54 | 62% | 74% | 9.26 | 2.93 | 1.79 | 68% | 81% |
| C06 | 1.85 | 0.41 | 0.40 | 78% | 78% | 1.91 | 0.67 | 0.74 | 65% | 61% |
| C07 | 3.42 | 0.34 | 0.36 | 90% | 89% | 3.09 | 1.33 | 1.18 | 57% | 62% |
| C08 | 3.41 | 0.38 | 0.38 | 89% | 89% | 3.35 | 1.76 | 0.65 | 47% | 81% |
| C09 | 2.47 | 0.33 | 0.34 | 87% | 86% | 2.43 | 0.70 | 0.57 | 71% | 77% |
| C10 | 3.48 | 0.78 | 0.77 | 78% | 78% | 3.21 | 1.01 | 0.99 | 68% | 69% |
| C11 | 1.63 | 1.28 | 1.12 | 21% | 32% | 1.87 | 1.59 | 1.51 | 15% | 19% |
| C12 | 3.38 | 7.14 | 3.24 | −111% | 4% | 3.37 | 2.96 | 2.92 | 12% | 14% |
| C14 | 1.77 | 4.61 | 0.93 | −161% | 48% | 3.74 | 3.37 | 3.39 | 10% | 9% |
| C16 | 1.87 | 0.80 | 0.67 | 57% | 64% | 1.83 | 0.89 | 0.87 | 51% | 53% |

Figure 10a,b shows the model one-day model prediction results of the G02 and G16 satellites, respectively. Figure 10a shows that both neural network models were able to model the error trends in the along-, cross-, and radial-track directions. As per the G02 satellite prediction results, the PSO–BP neural network achieved a better modeling effect in terms of the details of the error change in the along-track direction. Figure 10b shows that the two neural network models in the cross-track direction improved the orbit error from the meter level to the decimeter level. Figure 10c,d shows the three-day model prediction results. The two models achieved the best results in the C and R directions of the two satellites. In Figure 10c, due to the broadcast ephemeris orbit errors of the G02 satellite in the R direction being lower, the compensation effect of the two models is not obvious, but the PSO–BP neural network result curve is closer to 0 m than that of the BP neural network. The results of the two models in Figure 10d are roughly the same in the three directions of the G16 satellite.

Table 5 shows that in the GPS broadcast ephemeris orbit error modeling, the one-day STD and RMS in the along-track direction were reduced by 30–40%. Due to the large orbit error of the broadcast ephemeris in the radial-track direction of the G16 satellite, the one-day and three-day orbit error accuracies improved by 90% after modeling and compensation. Both neural network models were able to model the orbit error of GPS broadcast ephemeris and improve its orbit accuracy. In the three-day outcome, the two models were able to reduce the orbit error of the G02 and G16 satellites in the three directions, which is roughly the same as the one-day outcome.

Table 6 shows the results of the GPS satellites with no missing data. As shown in the one- and three-day results, the model output of more than 75% of the satellites could better compensate the broadcast ephemeris orbit errors. The results of the two models are not much different. The improvement rate of the orbit accuracy of most satellites was 30–50%, but the G23, G29, and G31 satellites demonstrated low accuracy and negative improvements. The three-day results and improvement rates of the two models changed little compared to the one-day results in most satellites.

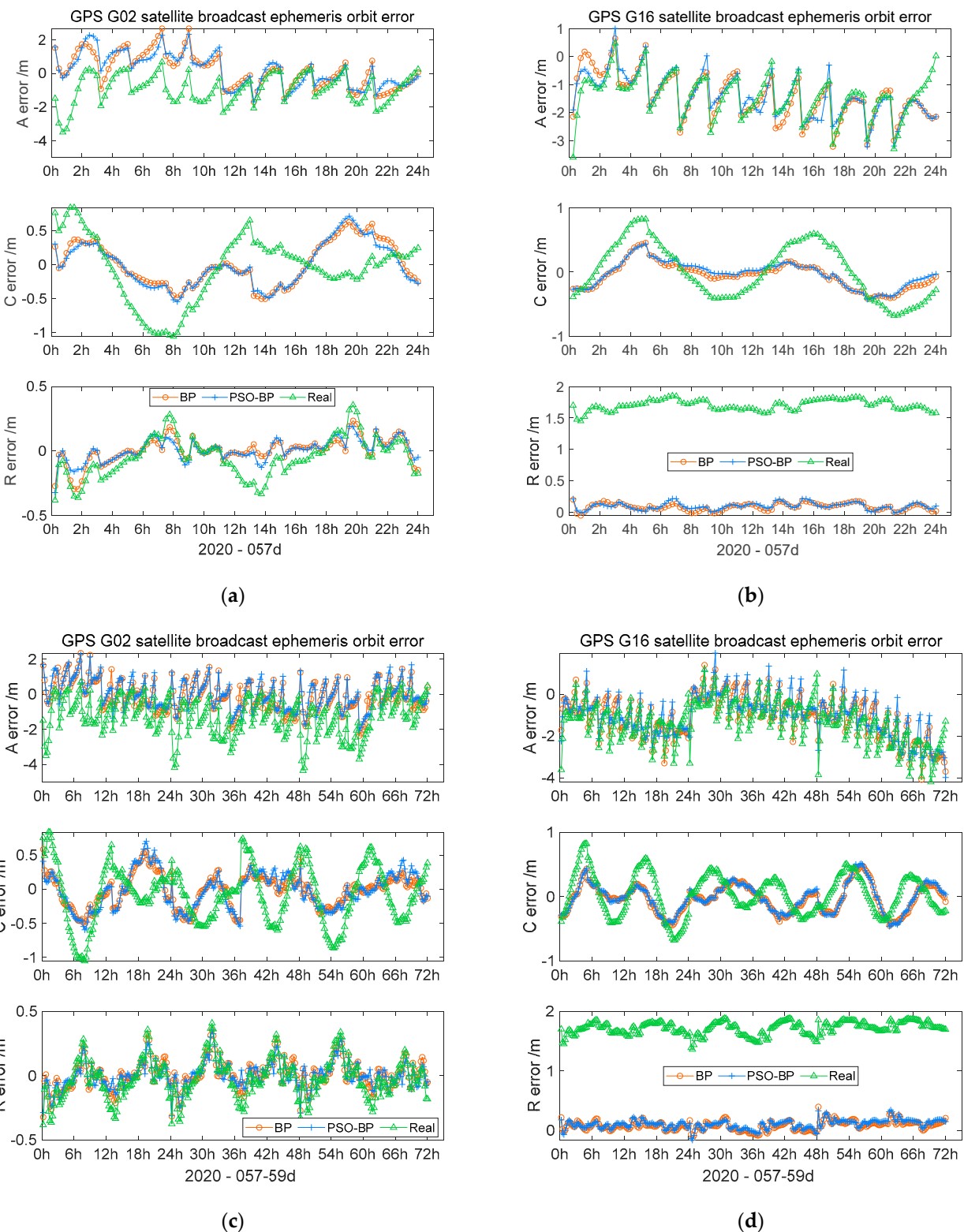

**Figure 10.** GPS G02/ G16 satellite orbit modeling outcome: (**a**) the satellite orbit modeling outcome of G02 on the 57th day of 2020; (**b**) the satellite orbit modeling outcome of G16 on the 57th day of 2020; (**c**) GPS G02 satellite orbit modeling outcome on the 57th–59th days of 2020; (**d**) GPS G16 satellite orbit modeling outcome on the 57th–59th days of 2020.

**Table 5.** G02/G16 satellite broadcast ephemeris and model-refined ephemeris orbit errors.

| Days | PRN | Direction | Along-Track | | | Cross-Track | | | Radial-Track | | |
|------|-----|-----------|------|------|------|------|------|------|------|------|------|
| | | Model | Mean/m | STD/m | RMS/m | Mean/m | STD/m | RMS/m | Mean/m | STD/m | RMS/m |
| 1 d | G02 | Real | −0.93 | 0.88 | 1.28 | −0.06 | 0.48 | 0.48 | −0.05 | 0.16 | 0.16 |
| | | BP | 0.17 | 1.02 | 1.03 | 0.01 | 0.31 | 0.31 | 0.01 | 0.10 | 0.10 |
| | | PSO–BP | 0.20 | 1.00 | 1.01 | −0.01 | 0.32 | 0.31 | 0.01 | 0.08 | 0.08 |
| | G16 | Real | −1.42 | 0.76 | 1.61 | 0.01 | 0.43 | 0.42 | 1.70 | 0.09 | 1.70 |
| | | BP | −1.42 | 0.82 | 1.64 | −0.06 | 0.20 | 0.21 | 0.09 | 0.06 | 0.10 |
| | | PSO–BP | −1.40 | 0.78 | 1.60 | −0.03 | 0.20 | 0.20 | 0.10 | 0.06 | 0.11 |
| 3 d | G02 | Real | −1.28 | 1.04 | 1.65 | −0.08 | 0.43 | 0.43 | −0.03 | 0.15 | 0.16 |
| | | BP | −0.12 | 0.85 | 0.86 | −0.02 | 0.24 | 0.24 | 0.01 | 0.11 | 0.11 |
| | | PSO–BP | 0.00 | 0.84 | 0.84 | 0.02 | 0.27 | 0.27 | 0.00 | 0.10 | 0.10 |
| | G16 | Real | −1.36 | 0.98 | 1.67 | 0.03 | 0.31 | 0.32 | 1.71 | 0.11 | 1.72 |
| | | BP | −1.10 | 0.99 | 1.48 | −0.03 | 0.22 | 0.22 | 0.09 | 0.08 | 0.11 |
| | | PSO–BP | −1.01 | 0.98 | 1.41 | −0.02 | 0.22 | 0.22 | 0.11 | 0.07 | 0.13 |

**Table 6.** GPS satellite broadcast ephemeris orbit error and model-refined ephemeris orbit errors of RMS(3D) and the improvement rate.

| PRN | RMS(3D)/m (1 Day) | | | Improvement Rate (1 Day) | | RMS(3D)/m (3 Days) | | | Improvement Rate (3 Days) | |
|-----|------|------|--------|------|--------|------|------|--------|------|--------|
| | Real | BP | PSO–BP | BP | PSO–BP | Real | BP | PSO–BP | BP | PSO–BP |
| G01 | 1.66 | 1.18 | 1.09 | 29% | 34% | 1.58 | 1.50 | 1.29 | 5% | 18% |
| G02 | 1.37 | 1.08 | 1.06 | 21% | 23% | 1.71 | 0.90 | 0.89 | 47% | 48% |
| G03 | 1.65 | 1.25 | 1.17 | 24% | 29% | 1.93 | 1.02 | 1.02 | 47% | 47% |
| G04 | 1.68 | 0.82 | 0.81 | 51% | 52% | 1.75 | 0.85 | 0.87 | 51% | 50% |
| G05 | 0.94 | 0.81 | 0.77 | 14% | 18% | 1.06 | 0.93 | 0.89 | 12% | 17% |
| G06 | 1.68 | 0.95 | 0.94 | 43% | 44% | 1.69 | 0.99 | 0.97 | 41% | 43% |
| G07 | 1.56 | 0.96 | 0.86 | 38% | 45% | 1.33 | 1.24 | 1.23 | 7% | 8% |
| G08 | 1.66 | 1.14 | 1.15 | 31% | 30% | 1.79 | 1.07 | 1.08 | 40% | 39% |
| G09 | 1.56 | 0.88 | 0.89 | 43% | 43% | 1.62 | 0.84 | 0.83 | 48% | 49% |
| G10 | 1.63 | 1.19 | 1.18 | 26% | 28% | 1.85 | 1.11 | 1.14 | 40% | 38% |
| G11 | 1.64 | 1.04 | 1.05 | 36% | 36% | 1.79 | 1.03 | 1.07 | 42% | 40% |
| G12 | 0.85 | 0.70 | 0.63 | 18% | 25% | 0.97 | 0.81 | 0.83 | 16% | 14% |
| G13 | 1.86 | 0.92 | 0.81 | 51% | 56% | 2.16 | 1.13 | 1.17 | 48% | 46% |
| G14 | 1.85 | 0.93 | 0.99 | 50% | 47% | 1.83 | 0.92 | 0.95 | 50% | 48% |
| G16 | 2.39 | 1.66 | 1.62 | 30% | 32% | 2.42 | 1.50 | 1.43 | 38% | 41% |
| G18 | 1.77 | 1.05 | 0.95 | 41% | 47% | 1.71 | 1.06 | 0.94 | 38% | 45% |
| G19 | 3.30 | 1.49 | 1.89 | 55% | 43% | 2.99 | 2.26 | 2.32 | 24% | 23% |
| G20 | 2.13 | 0.63 | 0.64 | 71% | 70% | 1.94 | 1.02 | 1.07 | 47% | 45% |
| G21 | 1.98 | 1.05 | 1.05 | 47% | 47% | 2.00 | 0.98 | 0.96 | 51% | 52% |
| G22 | 0.85 | 0.85 | 0.86 | 0% | −1% | 1.33 | 1.20 | 1.19 | 9% | 10% |
| G23 | 1.34 | 0.83 | 0.83 | 38% | 38% | 1.15 | 0.94 | 1.09 | 18% | 5% |
| G24 | 1.80 | 1.28 | 1.30 | 29% | 28% | 1.76 | 1.56 | 1.52 | 11% | 14% |
| G25 | 1.68 | 0.98 | 0.94 | 42% | 44% | 1.72 | 1.21 | 1.19 | 30% | 31% |
| G26 | 1.41 | 0.92 | 0.90 | 35% | 37% | 1.48 | 1.28 | 1.23 | 13% | 16% |
| G27 | 1.51 | 0.89 | 0.76 | 41% | 50% | 1.54 | 0.88 | 0.85 | 43% | 45% |
| G28 | 1.78 | 0.78 | 0.83 | 56% | 53% | 1.80 | 0.88 | 0.98 | 51% | 46% |
| G29 | 0.80 | 1.07 | 0.98 | −34% | −22% | 1.00 | 1.24 | 1.23 | −24% | −23% |
| G30 | 1.98 | 1.28 | 1.19 | 35% | 40% | 1.72 | 1.27 | 1.29 | 26% | 25% |
| G31 | 0.98 | 0.99 | 1.01 | −2% | −3% | 0.91 | 0.86 | 0.87 | 5% | 4% |

## 4. Discussion

In the GEO, IGSO, and MEO satellites of BDS-2, after eliminating the one- and three-day outputs of the two neural network models in the orbit error of the broadcast ephemeris, the orbit accuracy of broadcast ephemeris of most BDS-2 satellites could be improved. As

the output time of the model results increased, the three-day orbit accuracy of more than 70% of the satellites was slightly lower than that of the one-day results, but the degree of orbit accuracy loss of the PSO–BP neural network in the three-day outcome was lower than that of the BP neural network. The orbit accuracy of the IGSO and GEO satellites could be improved from 10 m to the decimeter level.

In the GEO satellites, the STD and RMS of the one- and three-day orbit error in the three directions were reduced by 20–70% in the PSO–BP neural network. The RMS(3D) of the PSO–BP neural network model in three directions in one day and three days was 20–30% higher than that of the BP neural network model. The IGSO satellite results showed that the PSO–BP neural network model output accuracy of the along- and radial-track directions achieved a 70–80% improvement in one and three days, and the one- and three-day improvement rate of RMS(3D) was 60–90% for most of the IGSO satellites, which is the same level as the BP neural network. Due to the occurrence of gross errors caused by the AODE, leading to poor model training, the BP neural network results of some of the MEO satellites were poor and unstable, but the PSO–BP neural network results of the RMS(3D) of all MEO satellites improved by 5–45% in the one-day and three-day outcomes. This shows that the PSO–BP neural network has a greater ability to resist gross errors than that of BP neural network for modeling the changing trend of broadcast ephemeris orbit errors accurately, along with more effectively improving the orbit accuracy of broadcast ephemeris.

In the GPS satellite orbit error model, both neural network models were able to model the broadcast ephemeris orbit error and improve the orbit accuracy of broadcast ephemeris of most satellites by 10–40% in the along-, cross-, and radial-track directions in one day, and one- and three-day improvement rates of RMS(3D) being 30–50%. With no AODE influence, the one- and three-day results in the GPS satellites were relatively consistent. However, some satellites had poor compensation accuracy and negative compensation. The reason is that more satellite errors lead to poor model training effects. Compared to the BP neural network, the PSO–BP neural network model experienced a smaller improvement in STD and RMS in the GPS orbit error results, but in the model output comparison chart, its result was closer to 0 m.

## 5. Conclusions and Suggestions

In this paper, BP and PSO–BP neural networks were used to model the orbit error of broadcast ephemeris. The experiments proved that both models can build an accurate model of orbit errors with good modeling effects on the change trend of broadcast ephemeris satellite orbit error.

The PSO–BP neural network model achieved better outcomes than BP neural network model in most cases; it could more accurately model the broadcast ephemeris orbit error and more significantly improve the broadcast ephemeris orbit accuracy.

In future research work, we will mainly focus on the preprocessing of broadcast ephemeris orbit errors, including data smoothing for abnormally abrupt epochs, minimizing the impact of sudden changes on the training neural network model, and further analyzing the impact of the AODE of broadcast ephemeris on the orbit error. Meanwhile, neural network modeling of the broadcast ephemeris clock error will be analyzed, and these results will be applied to improve GNSS single-point positioning.

**Author Contributions:** Conceptualization, H.C. and X.S.; methodology, H.C.; software, X.S.; validation, Z.L. and F.N.; formal analysis, X.S.; investigation, H.C.; resources, Q.L.; data curation, Q.L.; writing—original draft preparation, H.C.; writing—reviewing and editing, H.C. and T.G.; visualization, X.S.; supervision, X.S. and T.G.; project administration, X.S. and T.G.; funding acquisition, X.S. and T.G. All authors have read and agreed to the published version of the manuscript.

**Funding:** This work was supported by the National Natural Science Foundation of China (No. 41974036); Talent Introduction Plan for Youth Innovation Team in Universities of Shandong Province (innovation team of satellite positioning and navigation); the Key Laboratory of Geomatics and Digital Technology of Shandong Province; Shandong Provincial Natural Science Foundation, China

(grant number ZR2018PD006); the Scientific Research Foundation of Shandong University of Science and Technology for Recruited Talents (grant number 2017RCJJ072).

**Data Availability Statement:** The raw data of GNSS broadcast ephemeris in the paper were obtained from ftp://igs.gnsswhu.cn/pub/gnss/mgex/daily/rinex3/2020/, accessed on 14 November 2021. The GNSS precise orbit products were obtained from ftp://igs.gnsswhu.cn/pub/gnss/products/mgex/, accessed on 14 November 2021. The processing and results data presented in this study are uploaded to the database of Remote Sensing to be retrievable.

**Acknowledgments:** The authors would like to thank the IGS Multi-GNSS Experiment (MGEX), iGMAS for providing GNSS relevant data and products, all of which enable this study.

**Conflicts of Interest:** The authors declare no conflict of interest.

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
