# Peer review of "Initial Results of Modeling and Improvement of BDS-2/GPS Broadcast Ephemeris Satellite Orbit Based on BP and PSO-BP Neural Networks"

_remotesensing, doi:10.3390/rs13234801_

Round 1
Reviewer 1 Report
The paper uses BP neural network and PSO-BP neural network to model the orbit error of GPS and BDS broadcast ephemeris. It is an interesting topic and provides some valuable information to real time services performance of GNSS. The results and their associated analysis are demonstrative as well supportive to the conclusion drown. However, the method is not introduced clearly, Further comments please see the attachment。

Author Response
Dear Review,
Please see the attachment.
With kind regards,
Yours sincerely,
Xing Su

Reviewer 2 Report
Language improvement: Lowwer => lower
“Because BDS global ground stations are located in China”, global ground station? In China? wording errors !
Check other places as well…
Other comments
1.“Table 1. C03 satellite broadcast ephemeris orbit error” compares the refined ephemeris using two neural network models and the real broadcast ephemeris with precise orbits, I think the refined ephemeris does not belong to broadcast, suggest to change a name.
- One-day only data was used to validate the models, it seems to me not enough, could you please extend the analysis to longer time span?
Author Response

(The authors gave the same response as above.)

Reviewer 3 Report
Dear Authors,
This article deals with an estimation of orbit errors in broadcast orbits of BDS and GPS through two neural network methods. The method and results are well presented. I think this article has enough quality and contents to be published. However, I have some comments.
- The merit of the method in this article is not clear. Introduction says that coordinate, velocity, and epoch time are used as training samples. When you suggest a new method or idea, you have to show advantages of the method. And it needs more explanation about PSO-BP and BP methods.
- It needs more details or specific examples of application of this methods. A neural network can estimate an amount of error because the error has some pattern or characteristics that can be modeled. It also means the broadcast orbits are generated with these weakness. So, I think the result can be used to update the broadcast orbit generation model. Also you can analyze what is the main source of these errors, I mean characteristics kind of periodic.
- Authors choose 6 days of training data. Is there any specific reason? Is there any ideas if there is maneuver of eclipse? It would be very helpful to evaluate your methods.
Best Regards,
Author Response

(The authors gave the same response as above.)
